# Health-Related Quality of Life in Long-Term Colorectal Cancer Survivors

**DOI:** 10.3390/healthcare12191917

**Published:** 2024-09-25

**Authors:** Alba Marcos-Delgado, Vicente Martín-Sánchez, Ana Molina-Barceló, Jessica Alonso-Molero, Beatriz Pérez-Gómez, Marina Pollán, Nuria Aragonés, María Ederra-Sanza, Guillermo Fernández-Tardón, Gemma Binefa, Victor Moreno, Rocío Barrios-Rodríguez, Pilar Amiano, José María Huerta, Enrique Pastor Teso, Juan Alguacil, Gemma Castaño-Vinyals, Manolis Kogevinas, Antonio José Molina de la Torre

**Affiliations:** 1The Research Group in Gene-Environment and Health Interactions, Institute of Biomedicine (IBIOMED), Universidad de León, 24071 León, Spain; vmars@unileon.es (V.M.-S.); ajmolt@unileon.es (A.J.M.d.l.T.); 2Department of Biomedical Sciences, Area of Preventive Medicine and Public Health, Universidad de León, 24071 León, Spain; 3CIBER Epidemiology and Public Health CIBERESP, 28029 Madrid, Spain; alonsomoleroj@gmail.com (J.A.-M.); bperez@isciii.es (B.P.-G.); mpollan@isciii.es (M.P.); nuria.aragones@salud.madrid.org (N.A.); maria.ederra.sanz@navarra.es (M.E.-S.); fernandeztguillermo@uniovi.es (G.F.-T.); gbinefa@iconcologia.net (G.B.); v.moreno@iconcologia.net (V.M.); rbarrios@ugr.es (R.B.-R.); p-amiano@euskadi.eus (P.A.); jmhuerta.carm@gmail.com (J.M.H.); alguacil@dbasp.uhu.es (J.A.); gemma.castano@isglobal.org (G.C.-V.); manolis.kogevinas@isglobal.org (M.K.); 4Cancer and Public Health Area, FISABIO-Public Health, 46022 Valencia, Spain; ana.molina@fisabio.es; 5Department of Preventive Medicine and Public Health, IDIVAL, Universidad de Cantabria, 39005 Cantabria, Spain; 6Department of Epidemiology of Chronic Diseases, National Center for Epidemiology, Carlos III Institute of Health, 28029 Madrid, Spain; 7Epidemiology Section, Public Health Division, Department of Health of Madrid, 28035 Madrid, Spain; 8Navarra Public Health Institute, 31003 Pamplona, Spain; 9IdiSNA, Navarra Institute for Health Research, 31008 Pamplona, Spain; 10ISPA (Health Research Institute of the Pincipality of Asturias), IUOPA, University of Oviedo, 33006 Asturias, Spain; 11Cancer Prevention and Control Program, Catalan Institute of Oncology-IDIBELL, L’Hospitalet de Llobregat, 08901 Barcelona, Spain; 12Department of Clinical Sciences, Faculty of Medicine, University of Barcelona, 08036 Barcelona, Spain; 13Departamento de Medicina Preventiva y Salud Pública, Universidad de Granada, 18071 Granada, Spain; 14Instituto de Investigación Biosanitaria ibs, 18012 Granada, Spain; 15Public Health Division of Gipuzkoa, Biodonostia Research Institute, 20014 San Sebastian, Spain; 16Department of Epidemiology, Murcia Regional Health Council, IMIB-Arrixaca, 30120 Murcia, Spain; 17Servicio de Cirugía General y del Aparato Digestivo, Complejo Asistencial Universitario de León, 24008 León, Spain; epastort@saludcastillayleon.es; 18Centro de Investigación en Salud y Medio Ambiente (CYSMA), Universidad de Huelva, 21004 Huelva, Spain; 19Instituto de Salud Global de Barcelona (ISGlobal), 08036 Barcelona, Spain; 20Campus del Mar, Universitat Pompeu Fabra (UPF), 08002 Barcelona, Spain; 21IMIM (Hospital del Mar Medical Research Institute), 08003 Barcelona, Spain

**Keywords:** health-related quality of life (HRQoL), colorectal cancer, cancer survivors

## Abstract

The aim of our study is to evaluate the relationship between sociodemographic and clinical characteristics of individuals with Colorectal Cancer (CRC), tumour-intrinsic characteristics and treatment received with health-related quality of life (HRQoL). Methods: Cross-sectional analysis of data from 805 survivors from the MCC study was conducted. HRQoL was assessed through a general and specific questionnaire, SF-12 and FCSI (Colorectal Symptom Index). Statistical analyses were performed with linear regression with adjustment for sociodemographic variables, stage at diagnosis and histological grade. Results: Participants had survived a median of 7.9 years from diagnosis (IQR 7.1–8.5 years). Age at diagnosis, sex and area showed a clear association with HRQoL in both physical and mental dimensions of the SF-12 questionnaire. A direct association between CRC recurrence was also found in the PCS-12 and MCS-12 dimensions and radical surgery in the PCS-12. Regarding the scores in FCSI questionnaire, statistically significant differences were observed by sex, age and area, with older women being the most impaired (*p* < 0.001). Conclusions: Age, sex and area was associated with lower scores of HRQoL among CRC survivors. Knowing the determinants related to HRQoL would allow us to lay the groundwork to develop strategies that help reduce morbidity and mortality, relapses and increase HRQoL.

## 1. Introduction

Colorectal cancer (CRC) is a major public health problem. Worldwide, nearly 2 million new cases are diagnosed each year, and more than 900,000 deaths are caused by CRC [1,2]. The distribution of CRC incidence represents a large geographical variability, with the highest rates corresponding to developed countries. In Spain, it is the cancer with the highest incidence and the second highest in mortality, with a growing trend [3,4]. The 5-year survival rate in our country has improved from 56.5% and 55.2% for the colon and rectum in 2000–2004 to 63.2% and 59.5%, respectively in 2010–2014, so that the prevalent cases of five-year survivors are estimated at more than 100,000. The trend in the incidence of CRC, the ageing of the population, screening programmes and improvements in treatment augur a significant increase in the number of CRC survivors in our country [3,5]. Many researchers are analysing health-related quality of life (HRQoL) in cancer survivor patients, and while it appears to be decreased in these patients [6,7], it is not so clear which spheres are affected, and whether over time HRQoL equals that of other individuals without that pathology. In Spain, those studies that have attempted to learn about HRQoL in survivors of CRC are very limited and with very small samples [8].

The main factors associated with changes in HRQoL include sociodemographic variables such as sex, age, marital status, occupation and educational level; clinical and treatment related variables; and other lifestyle-related factors such as body mass index (BMI) and social support [7,9,10,11,12,13]. In addition, HRQoL is a strong predictor of long-term disability and mortality. Poor HRQoL can increase the likelihood of complications or disease recurrence. For instance, patients with low HRQoL may experience heightened stress level, impaired immune function, and diminished physical and mental well-being. Additionally, reduced quality of life is associated with lower adherence to medical treatment, which can directly impact survival rates [14,15,16]. However, there are few long-term studies conducted on CRC survivors [6,7,8]. Knowing the determinants related to HRQoL would allow us to lay the groundwork to develop strategies that help reduce morbidity and mortality, reduce relapses and increase HRQoL. Therefore, the main objective of our study is to evaluate the relationship between sociodemographic and clinical characteristics of individuals with CRC, tumour-intrinsic characteristics and treatment received with HRQoL measured using SF-12 (12-Item Short-Form Health Survey) and FCSI (Colorectal Symptom Index).

## 2. Materials and Methods

### 2.1. Study Design and Participants

The MCC-Spain study recruited controls and colorectal cancer patients between 2008 and 2013 [17]. The cases have become a follow-up cohort for which information is available until 2017–2018. There are a total of 2140 cases of initial CRC at various stages of the disease, of which 2094 have been followed up, accumulating 12,813 patient-years of follow-up, and 1230 were alive at the time of follow-up. A total of 805 cases, including 486 men and 319 women, were included in the final sample, with a 65.4% response rate (Figure 1).

At the time of recruitment, information was available on socio-demographic characteristics, risk factors for CRC, stage, classification, molecular biomarkers and first-line treatment. Participants had survived a median of 7.9 years from diagnosis (IQR 7.1–8.5 years), and the average age of the studied population was 65.1 (10.2).

Follow-up was conducted between 2017 and 2018 by reviewing medical records and collecting or updating information on complete clinical remission, treatment response, and current patient living status, among others. The vital state of patients without a consultation in the last three months was ascertained by consulting the National Death Index. Subsequently, living patients were contacted by telephone and asked to complete two quality of life questionnaires: the SF-12 [18,19] and FCSI [20].

### 2.2. Ethical Considerations

The MCC-Spain protocol was approved by the ethics committee of all participating institutions. All participants were informed about the purpose of the study and gave their informed consent in writing. It included the authorization for following up with the patient via medical records or phone calls. Only participants agreeing to be followed up were included in this analysis. Confidentiality of information was ensured through a double registration system. The project’s databases are registered within the Data Protection Agency under number 2102672171.

### 2.3. Health-Related Quality of Life

The HRQoL was measured with the validated Spanish version of the SF-12 questionnaire [19,21,22], widely used as an accurate way to measure self-perceived HRQoL. Additionally, it has high internal consistency (Cronbach’s α > 0.70) [19]. This questionnaire consisted of 12 items that were grouped into 8 scales and in turn were assessed in two aggregate dimensions: the physical component summary scores (PCS) composed of physical functioning (PF), role-physical (RP), bodily pain (BP) and general health (GH) and mental component summary scores (MCS) composed of vitality (VT), social functioning (SF), role-emotional (RE) and mental health (MH). Scores were standardized for the physical and mental components; a higher score means a better HRQoL.

The other questionnaire that was used to measure HRQoL was FCSI for patients with CRC, a FACT-Colorectal Symptom Index (a subset of FACT-C containing 9 items) [20,23]. It is a validated questionnaire with high internal consistency (Cronbach’s α 0.81) [23]. This questionnaire is a brief, symptom-specific measure composed of 9 items that assess gastrointestinal symptoms, pain and satisfaction with current quality of life, measured on a Likert scale (0–4).

For the calculation of the SF-12 questionnaire score, first the indicator variables were created for each of the response categories of the items, then the aggregate scores were calculated by the sum of the weighted variables, then they were standardized to obtain a mean of 50 and sd of 10 and finally were calculated by the specific method with the weights corresponding to Spain [19]. The FCSI score was calculated as the sum of all item values, multiplied by nine, and divided by the number of items answered; each item was scored from 0 to 4. This FCSI score can have values between 0 and 36 [23]. A higher score means a better HRQoL in both questionnaires.

### 2.4. Study Variables

#### 2.4.1. Independent Variables

Sociodemographic characteristics included age at diagnosis (continuous and categorical variable), sex (men/women), province of recruitment (eleven Spanish regions), family history of colon cancer (none and some degree family history of colon cancer), education level (lower than primary or primary, secondary and university), civil status (single, married and widow), smoking (non-smoker and former at diagnosis/current smoker at diagnosis), and BMI was calculated as weight in kg divided by squared height in m. Participants were classified as “normal bmi” if <25 kg/m^2^, “overweight” if <30 kg/m^2^ and “obese” if >30 kg/m^2^.

Regarding tumour characteristics, these includ tumour size (T0-T4), node infiltration (N0-N2), metastasis (M0–M1), complete clinical remission (Yes/No), recurrence (Yes/No), TNM pathological stage (0–IV), histological grade (well, moderately and poorly differentiated) and histological type (Adenocarcinoma/Others).

Concerning variables related to treatment received at diagnosis, we included chemotherapy (Yes/No), radiotherapy (Yes/No) and surgery type (radical/palliative).

#### 2.4.2. Dependent Variables

The dependent variable was the HRQoL measured through the SF-12 questionnaire (PCS and MCS) and FCSI questionnaire.

### 2.5. Statistical Analysis

The final sample size in the study was determined by the number of individuals with colorectal cancer recruited according to the calculation proposed for the MCC-Spain case-control study, the number of deaths occurring in the study group during the follow-up period and the participation rate achieved in the telephone surveys carried out. Therefore, what has been assessed is that with the available sample size, there is a capacity to detect by means of multiple linear regression models of up to 25 variables (considering an alpha error of 0.05 and beta error of 0.2) an f2 value of effect size measurements of 0.03, which would correspond to R2 values of 0.028; so it seems an adequate sample size for the analyses proposed.

For sample description, means and standard deviations for quantitative variables were calculated, as well as the percentages for qualitative variables, consistent with the total sample and stratified by sex. Comparisons were made by sex in the descriptive characteristics of the sample using *t*-test or χ^2^ test as appropriate.

In this case-only study, the associations between HRQoL scores with independent variables were examined with linear regression model adjusted for age at diagnosis, sex, educational level, province of recruitment, stage at diagnosis and histological grade. The different categories of the variables used in the linear regression models were based on previous studies published [24]. In addition, stratified analyses by sex were carried out. The results are presented as marginal means and the Storey–Tibshirani method was used as a correction test [25].

All statistical analyses were done using the StataCorp statistical package. 2019. Stata Statistical Software: Release 16.1. College Station, TX, USA: StataCorp LLC. We considered a two-tailed value of 0.05 to be the threshold for statistical significance.

## 3. Results

A total of 805 cases, including 486 men and 319 women, were included in the final sample, with a 65.4% response rate (Figure 1).

Participants had survived a median of 7.9 years from diagnosis (IQR 7.1–8.5 years). The descriptive characteristics of the sample are shown in Appendix A. The average age of the studied population was 65.1 (10.2); 81.5% were married, and 66.6% had primary education or less. Regarding the descriptive characteristics of the tumour and treatment received at diagnosis, 56.0% of the studied population had a tumour size of T3, 67.0% had a N0 node infiltration, 93.9% had no metastasis, 58.0% received chemotherapy, 24.1% radiotherapy and 91.9% received radical surgery. There was a significant difference between males and females with respect to sociodemographic variables and treatment received at diagnosis. Women were older and with a lower education level, included fewer smokers, had a lower BMI and lower quality of life than men (Table 1).

Age at diagnosis, sex and area showed a clear association with HRQoL in both physical and mental dimensions of the SF-12 questionnaire (Figure 1). Women presented lower scores than men in both spheres (Appendix A). As regards age, lower scores were observed among patients with CRC diagnosis after 76 years. A direct association between CRC recurrence was also found in the PCS-12 and MCS-12 dimensions and radical surgery in the PCS-12. On the other hand, in men lower scores were found between the physical dimension and being a smoker at diagnosis (Table 2 and Table 3).

Regarding the scores in the specific questionnaire of HRQoL in CRC (FCSI), statistically significant differences were observed by sex, age and area, with older women being the most impaired (Figure 2). Also, smokers reported lower scores, as did men with recurrence in the tumour (Table 4). Finally, no association between FCSI and treatment received was observed in both sexes (Appendix A).

## 4. Discussion

Our study is composed of a large sample of CRC survivors (more than 7 years since diagnosis) with a heterogeneous geographical location throughout Spain.

The data analysed in our study suggest that age and sex are the variables that most determine the HRQoL of patients with long survival of CRC. Regarding age, scientific evidence is controversial; in most studies it is the younger patients who are more affected in the mental dimensions [26]. This may be because they have fewer coping strategies and resources needed to manage a potentially life-threatening disease such as CRC [27]. However, our results are consistent with the study published by Jansen et al. [7], with younger patients being the most affected in the physical dimensions, while in the mental sphere it is the older patients who suffer the most. They assessed HRQoL 1, 3, 5 and 10 years after diagnosis in a population-based cohort in Germany.

No association was found between quality of life and tumour characteristics and treatment received in sex-stratified analyses beyond recurrence and surgery type; this may be due, on the one hand, to the lower survival rate in patients diagnosed at a higher stage and, on the other hand, to the concept of benefit-finding (BF), defined as reported benefits resulting from trauma, illness or other negative experiences. BF is considered a form of appraisal, as it involves a selective cognitive process used to evaluate a situation, reducing the sense of victimization by focusing on positive aspects that help individuals adapt [28,29]. In addition, through this concept it is possible to explain the difference in the association between HRQoL and recurrence between men and women, since according to a study published by Rinaldis et al. it is women who perceive greater benefits than men as they perform a more positive reappraisal, which leads to an increase in BF. In the case of TNM pathological stage, no association has been found; this may be due to the lower survival rate in patients diagnosed at a higher stage because in our sample there is an under-representation of stage IV patients, who may report higher scores in the HRQoL [30].

Overall, our results reflect a lower mean score in the physical sphere compared to the mental sphere, which could be due to underlying comorbidities and residual symptoms [31,32]. It is also noteworthy that it is women who have worse scores in all scales and in both questionnaires compared to men. This is something widely described in the scientific literature, both in the general population, in CRC and in other types of cancer [13,33,34]. Previous studies suggest that patients with CRC of long evolution (>5 years) obtain similar scores to the general population according to age and sex ranges [35]. In our case, if we compare the scores of the SF-12 questionnaire between survivors and the general Spanish population [19], it shows how the mental sphere received worse results in those diagnosed with cancer, while the physical sphere is more affected in the general population. This contrasts with a systematic review analysing 10 studies assessing HRQoL among long-term (>5 years) colorectal cancer survivors, in which although survivors indicated a good overall QoL, it was the physical domains that they scored worse than the general population, with these differences being associated with age, sex, obesity, smoking, socioeconomic status and other comorbidities [26]. However, a recently published study by Pate et al. [36] shows that scores on questionnaires assessing overall HRQoL are similar in both groups. Moreover, no significant differences have been found in the scores obtained through the questionnaire for the general population (SF-12) and the one specific to CCR (FCSI). This may be because, as we mentioned earlier, the HRQoL of long-term (>5 years) CRC survivors may resemble that of the general population.

Our study has several strengths including the sample size, the inclusion of 11 geographically diverse sites of data collection and use of validated cancer-specific and general assessments to measure HRQoL. In addition, the study of HRQOL with a gender perspective in long-term survivors of CRC has important applications in clinical practice, being able to identify the variables that most affect the patient at the time of diagnosis and to intervene in them.

One of the main limitations of this study is that it is a case-only cross-sectional study, where we have no prior measurement of HRQoL at the time of diagnosis. On the other hand, the questionnaire could present limitations inherent in them. In addition, telephone interviews may show an increase in scores on HRQoL; however, by conducting telephone interviews we ensure that the participant has understood the question. Another limitation may be the missing data, as individuals who did not respond to the questionnaires may have a lower quality of life; however, we have achieved a high response rate.

## 5. Conclusions

In summary, an association was found between age, sex and area and HRQOL in long-term colorectal cancer survivors. Women have lower scores on both the general HRQOL questionnaire SF-12 and CRC specific questionnaire FCSI. In terms of tumour characteristics, we found a decrease in scores in patients with recurrence in both questionnaires and radical surgery in physical dimensions. According to the treatment received, there were no significant differences between those who had undergone chemotherapy and radiotherapy.

## Figures and Tables

**Figure 1 healthcare-12-01917-f001:**
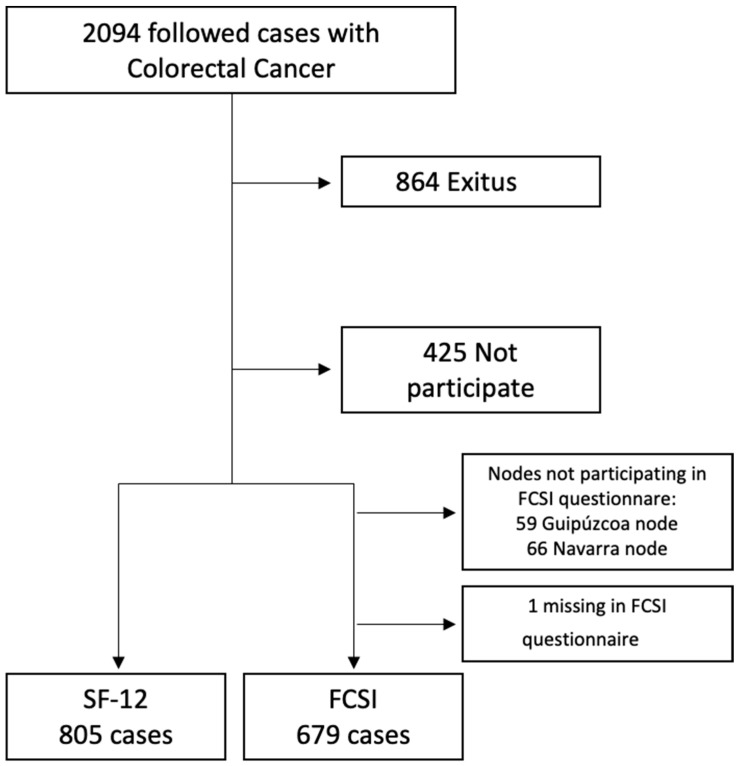
Flow chart of the followed cases of Colorectal Cancer in MCC study.

**Figure 2 healthcare-12-01917-f002:**
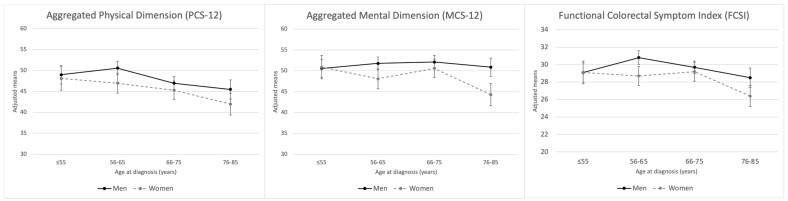
Marginal means for aggregated physical/mental dimensions of health-related quality of life and functional colorectal symptom index by baseline categories of age at diagnosis, stratified by sex.

**Table 1 healthcare-12-01917-t001:** Description of the sociodemographic variables, tumour characteristics and treatment received at diagnosis according to men or women.

		Quality of Life Questionnaire (*n* (%))		
		Men	Women	
		*n =* 486	*n =* 319	*p*-Value
Age at diagnosis	mean (sd)	64.9 (9.8)	65.4 (10.8)	<0.001
Age	≤55 *	84 (17.3)	68 (21.3)	0.01
	56–65	171 (35.2)	84 (26.3)	
	66–75	154 (31.7)	96 (30.1)	
	76–85	77 (15.8)	71 (22.3)	
Area	Asturias *	14 (2.9)	12 (3.8)	0.39
	Barcelona	125 (25.7)	87 (27.3)	
	Cantabria	33 (6.8)	28 (8.8)	
	Granada	15 (3.1)	15 (4.7)	
	Guipúzcoa	38 (7.8)	21 (6.6)	
	Huelva	21 (4.3)	13 (4.1)	
	León	107 (22.0)	63 (19.8)	
	Madrid	54 (11.1)	40 (12.5)	
	Murcia	5 (1.0)	2 (0.6)	
	Navarra	49 (10.1)	17 (5.3)	
	Valencia	25 (5.1)	21 (6.6)	
Education level	Primary education or less *	310 (63.8)	226 (70.9)	0.11
	Secondary education	120 (24.7)	62 (19.4)	
	University	56 (11.5)	31 (9.7)	
Civil status	Single *	24 (4.9)	26 (8.2)	<0.001
	Married or Living with a partner	445 (91.6)	211 (66.1)	
	Widow	17 (3.5)	82 (25.7)	
Smoking	Non-smoker and former at diagnosis *	123 (25.3)	233 (73.0)	<0.001
	Smoker at diagnosis	363 (74.7)	86 (27.0)	
BMI	17.5–25 *	125 (25.7)	129 (40.4)	<0.001
	25–29.9	235 (48.4)	124 (38.9)	
	≥30	126 (25.9)	66 (20.7)	
Family history of colon cancer	None *	336 (69.1)	206 (64.6)	0.18
	Some degree family history of colon cancer	150 (30.9)	113 (35.4)	
Tumour size	T0 *	33 (6.8)	19 (6.0)	0.91
	T1	35 (7.2)	26 (8.2)	
	T2	87 (17.9)	49 (15.4)	
	T3	270 (55.6)	181 (56.7)	
	T4	47 (9.7)	33 (10.3)	
	Missing	14 (2.9)	11 (3.5)	
Node infiltration	N0 *	324 (66.7)	215 (67.4)	0.94
	N1	109 (22.4)	67 (21.0)	
	N2	37 (7.6)	27 (8.5)	
	Missing	16 (3.3)	10 (3.1)	
Metastasis	M0 *	451 (92.8)	305 (95.6)	0.11
	M1	25 (5.1)	7 (2.2)	
	Missing	10 (2.1)	7 (2.2)	
Complete clinical remission	No *	481 (99.0)	319 (100)	0.07
	Yes	5 (1.0)	0	
	Missing	0	0	
Recurrence	No *	441 (90.7)	302 (94.7)	0.04
	Yes	45 (9.3)	17 (5.3)	
	Missing	0	0	
TNM pathological stage	0 *	26 (5.4)	16 (5.0)	0.48
	I	97 (20.0)	65 (20.4)	
	II	183 (37.7)	125 (39.2)	
	III	130 (26.8)	89 (27.9)	
	IV	25 (5.1)	7 (2.2)	
	Missing	25 (5.1)	17 (5.33)	
Histological grade	Well differentiated *	155 (31.9)	94 (29.5)	0.72
	Moderately differentiated	237 (48.8)	168 (52.7)	
	Poorly differentiated	48 (9.9)	27 (8.5)	
	Missing	46 (9.5)	30 (9.4)	
Histological type	Adenocarcinoma *	446 (91.8)	291 (91.2)	0.79
	Other	24 (4.9)	18 (5.6)	
	Missing	16 (3.3)	10 (3.1)	
Chemotherapy	No *	176 (36.2)	149 (46.7)	0.004
	Yes	299 (61.5)	168 (52.7)	
	Missing	11 (2.3)	2 (0.6)	
Radiotherapy	No *	322 (66.3)	241 (75.6)	0.02
	Yes	132 (27.2)	62 (19.4)	
	Missing	32 (6.6)	16 (5.0)	
Surgery Type	Radical *	448 (92.2)	292 (91.5)	0.76
	Palliative	20 (4.1)	12 (3.8)	
	Missing	18 (3.7)	15 (4.7)	
SF-12	PCS-12 (mean (sd))	48.4 (10.3)	45.6 (11.0)	<0.001
	MCS-12 (mean (sd))	51.5 (9.7)	48.6 (11.4)	<0.001
FCSI	mean (sd)	29.8 (4.7)	28.4 (4.9)	<0.001

* Categories used as reference in the analysis.

**Table 2 healthcare-12-01917-t002:** Marginal means of physical component summary (PCS-12) according to sociodemographic variables, tumour characteristics and treatment received at diagnosis.

		Physical Component Summary (PCS-12)					
		Total Sample		Men		Women	
		(*n* = 805)		(*n* = 486)		(*n* = 319)	
		Mean (95% CI)	*p*-Value (q-Values)	Mean (95% CI)	*p*-Value (q-Value)	Mean (95% CI)	*p*-Value (q-Value)
Sex	Men *	48.1 (45.1–53.1)	<0.001 (0.007)				
	Women	45.7 (44.6–46.9)					
Age at diagnosis	beta (95% CI)	−0.19 (−0.27–(−0.12))	<0.001 (<0.001)	−0.18 (−0.31–(−0.06))	0.004 (0.003)	−0.18 (−0.31–(−0.06))	0.004 (0.021)
Age	≤55 *	48.9 (47.2–50.6)	<0.001 (<0.001)	49.0 (46.8–51.2)	0.001 (0.009)	48.1 (45.3–50.9)	0.01 (0.006)
	56–65	49.3 (48.0–50.6)		50.6 (49.0–52.1)		47.0 (44.6–49.4)	
	66–75	46.3 (45.0–47.5)		47.0 (45.4–48.6)		45.3 (43.1–47.6)	
	76–85	43.9 (42.2–45.6)		45.5 (43.2–47.8)		42.0 (39.4–44.7)	
Area	Asturias *	49.1 (45.1–53.1)	0.003 (0.02)	47.5 (42.2–52.8)	0.003 (0.02)	50.7 (44.3–57.0)	0.53 (0.51)
	Barcelona	47.7 (46.3–49.2)		49.1 (47.2–52.8)		45.8 (43.3–48.2)	
	Cantabria	48.5 (45.9–51.2)		50.0 (46.5–53.5)		46.7 (42.5–50.8)	
	Granada	43.4 (45.9–51.2)		44.6 (39.4–49.8)		42.7 (37.1–48.3)	
	Guipúzcoa	47.6 (44.9–50.4)		47.7 (44.4–51.0)		47.7 (42.8–52.6)	
	Huelva	42.6 (39.1–47.1)		42.3 (37.9–46.6)		43.5 (37.4–48.3)	
	León	48.9 (47.3–50.5)		50.1 (48.2–52.1)		46.7 (43.9–49.5)	
	Madrid	46.5 (44.3–48.6)		47.7 (45.0–50.5)		44.3 (40.9–47.8)	
	Murcia	38.5 (30.8–46.2)		39.0 (30.2–47.9)		37.6 (22.4–52.8)	
	Navarra	47.8 (45.2–50.3)		50.3 (47.4–53.2)		42.5 (37.2–47.8)	
Smoking	Non-smoker and former at diagnosis *	48.0 (46.8–49.2)	0.13 (0.29)	49.9 (48.1–51.7)	0.06 (0.18)	45.8 (44.3–47.2)	0.76 (0.58)
	Smoker at diagnosis	46.7 (45.7–47.7)		47.8 (46.8–48.9)		45.2 (42.6–47.9)	
	Yes	43.9 (41.3–46.5)		43.9 (40.9–46.8)		45.4 (40.0–50.7)	
	Missing	-		-		-	
Surgery Type	Radical *	47.0 (46.3–47.8)	0.01 (0.06)	48.1 (47.2–49.0)	0.13 (0.29)	45.5 (44.2–46.7)	0.20 (0.36)
	Palliative	52.9 (49.1–56.7)		52.8 (48.0–57.5)		51.4 (44.7–58.1)	

Results adjusted for age at diagnosis, sex, educational level, province of recruitment, stage at diagnosis and histological grade at diagnosis analysis. * Categories used as reference in the analysis. (1) PCS-12: Physical Component Summary of SF-12, (2) MCS-12: Mental Component Summary of SF-12, (3) FCSI: a (Functional Assessment of Cancer Therapy) Colorectal Symptom Index.

**Table 3 healthcare-12-01917-t003:** Marginal means of mental component summary (MCS-12) according to sociodemographic variables, tumour characteristics and treatment received at diagnosis.

		Mental Component Summary (MCS-12)					
		Total Sample		Men		Women	
		(*n* = 805)		(*n* = 486)		(*n* = 319)	
		Mean (95% CI)	*p*-Value (q-Value)	Mean (95% CI)	*p*-Value (q-Value)	Mean (95% CI)	*p*-Value (q-Value)
Sex	Men *	51.5 (50.5–52.4)	<0.001				
	Women	48.8 (47.6–49.9)	(0.004)				
Age at diagnosis	beta (95% CI)	−0.04 (−0.11–0.04)	0.33 (0.58)	0.05 (−0.04–0.14)	0.29 (0.30)	−0.18	0.007 (0.01)
Age	≤55 *	50.7 (49.1–52.4)	0.02 (0.06)	50.6 (48.5–52.7)	0.63 (0.53)	50.9 (48.1–53.7)	0.002 (0.01)
	56–65	50.5 (49.2–51.7)		51.8 (50.3–53.3)		48.1 (45.7–50.5)	
	66–75	51.5 (50.2–52.7)		52.1 (50.5–53.6)		50.6 (48.4–52.9)	
	76–85	48.1 (46.4–49.8)		50.9 (48.7–53.1)		44.3 (41.6–47.0)	
Area	Asturias *	47.8 (43.8–51.7)	<0.001	49.4 (44.4–54.5)	0.01 (0.04)	44.2 (37.8–50.7)	0.001 (0.008)
	Barcelona	49.3 (47.9–50.7)	(<0.001)	50.0 (48.3–51.7)		48.5 (46.0–50.9)	
	Cantabria	53.2 (50.6–55.8)		52.8 (49.5–56.2)		52.9 (48.6–57.1)	
	Granada	46.6 (42.9–50.3)		48.4 (43.5–53.4)		44.8 (39.1–50.5)	
	Guipúzcoa	51.3 (48.6–54.0)		52.1 (49.0–55.2)		50.3 (45.3–55.3)	
	Huelva	43.5 (40.0–47.0)		45.7 (41.6–49.9)		39.6 (33.4–45.8)	
	León	52.5 (51.0–54.1)		53.3 (51.5–55.2)		51.5 (48.7–54.4)	
	Madrid	48.6 (46.5–50.7)		51.2 (48.6–53.8)		44.6 (41.1–48.1)	
	Murcia	44.9 (37.3–52.4)		47.7 (39.2–56.1)		39.1 (23.6–54.5)	
	Navarra	50.9 (48.4–53.4)		52.8 (50.0–55.5)		47.5 (42.1–52.8)	
	Valencia	55.2 (52.2 (58.2)		56.1 (52.3–59.9)		53.4 (48.4–58.3)	
Education level	Primary education or less *	49.9 (49.0–50.8)	0.22 (0.38)	50.8 (49.7–51.9)	0.09 (0.26)	48.6 (47.1–50.1)	0.40 (0.44)
	Secondary education	51.4 (49.9–53.0)		52.5 (50.8–54.3)		49.8 (46.9–52.7)	
	University	51.0 (48.8–53.2)		53.4 (50.8–55.9)		46.4 (42.3–50.5)	
Recurrence	No *	50.5 (49.8–51.3)	0.13 (0.29)	51.8 (50.9–52.7)	0.04 (0.13)	48.6 (47.3–49.8)	0.77 (0.58)
	Yes	48.5 (45.9–51.0)		48.7 (45.9–51.5)		49.4 (44.0–54.8)	

Results adjusted for age at diagnosis, sex, educational level, province of recruitment, stage at diagnosis and histological grade at diagnosis analysis. * Categories used as reference in the analysis.

**Table 4 healthcare-12-01917-t004:** Marginal means of functional assessment of cancer therapy colorectal symptom index (FCSI) according to sociodemographic variables, tumour characteristics and treatment received at diagnosis.

		Functional Assessment of Cancer Therapy Colorectal Symptom Index (FCSI)					
		Total Sample		Men		Women	
		(*n* = 679)		(*n* = 398)		(*n* = 281)	
		Mean (95% CI)	*p*-value (q-value)	Mean (95% CI)	*p*-value (q-value)	Mean (95% CI)	*p*-value (q-value)
Sex	Men *	29.7 (29.3–30.2)	<0.001				
	Women	28.4 (27.9–29.0)	(0.004)				
Age at diagnosis	beta (95% CI)	−0.05 (−0.08–0.56)	0.01 (0.05)	−0.03 (−0.08–0.01)	0.16 (0.32)	−0.07 (−0.13–(−0.01))	0.02 (0.07)
Age	≤55 *	29.1 (28.2–29.9)	<0.001	29.1 (28.0–30.2)	0.005 (0.02)	29.1 (27.8–30.4)	0.003 (0.02)
	56–65	30.0 (29.4–30.7)	(0.002)	30.8 (30.0–31.6)		28.7 (27.6–29.8)	
	66–75	29.5 (28.9–30.1)		29.7 (29.0–30.5)		29.2 (28.1–30.2)	
	76–85	27.6–26.8–28.4)		28.5 (27.4–29.6)		26.4 (25.2–27.6)	
Area	Asturias *	29.4 (27.6–31.2)	<0.001	29.7 (27.8–32.1)	0.001	28.6 (25.8–31.4)	0.01 (0.06)
	Barcelona	28.8 (28.2–29.5)	(<0.001)	29.4 (28.5–30.2)	(0.009)	28.0 (27.0–29.1)	
	Cantabria	30.0 (28.8–31.2)		30.5 (28.9–32.1)		29.2 (27.3–31.0)	
	Granada	27.0 (25.3–28.6)		27.0 (24.6–29.4)		27.1 (24.7–29.6)	
	Guipúzcoa	-		-		-	
	Huelva	26.0 (24.4–27.6)		26.2 (24.2–28.3)		25.9 (23.2–28.6)	
	León	30.5 (29.8–31.2)		30.8 (29.9–31.7)		30.2 (29.0–31.4)	
	Madrid	28.8 (27.9–29.8)		30.3 (29.0–31.5)		26.8 (25.2–28.3)	
	Murcia	26.5 (23.0–29.9)		26.6 (22.6–30.7)		26.5 (19.8–33.2)	
	Navarra	-		-		-	
	Valencia	30.0 (28.6–31.4)		30.4 (28.6–32.2)		29.2 (27.0–31.3)	
Civil status	Single *	28.1 (26.7–29.5)	0.24 (0.38)	27.6 (25.4–29.7)	0.10 (0.27)	28.3 (26.3–30.3)	0.89 (0.62)
	Married or Living with a partner	29.3 (28.9–29.7)		29.8 (29.4–30.3)		28.5 (27.8–29.2)	
	Widow	29.0 (27.9–30.1)		30.7 (28.3–33.1)		28.2 (27.0–29.4)	
Smoking	Non-smoker and former at diagnosis *	29.8 (29.2–30.3)	0.02 (0.07)	30.4 (29.5–31.3)	0.14 (0.30)	28.7 (28.1–29.4)	0.10 (0.27)
	Smoker at diagnosis	28.7 (28.2–29.3)		29.6 (29.1–30.1)		27.4 (26.1–28.7)	
Metastasis	M0 *	29.1 (28.7–29.5)	0.07 (0.21)	29.6 (29.2–30.1)	0.08 (0.23)	28.3 (27.8–28.9)	0.57 (0.53)
	M1	30.8 (29.0–32.7)		31.4 (29.3–33.6)		30.1 (25.8–34.4)	
Recurrence	No *	28.4 (29.0–29.8)	<0.001	30.0 (29.6–30.5)	0.005 (0.005)	28.5 (27.9–29.1)	0.23 (0.38)
	Yes	27.0 (25.8–28.3)	(0.004)	27.3 (25.9–28.8)		26.9 (24.4–29.4)	
Radiotherapy	No *	29.5 (29.0–29.9)	0.11 (0.28)	30.0 (29.5–30.6)	0.33 (0.41)	28.7 (28.0–29.3)	0.08 (0.23)
	Yes	28.6 (27.8–29.3)		29.3 (28.5–30.2)		27.1 (25.8–28.4)	
Surgery Type	Radical *	29.1 (28.7–29.4)	0.06 (0.19)	29.7 (29.2–30.1)	0.35 (0.43)	28.3 (27.7–28.8)	0.26 (0.38)
	Palliative	30.9 (29.2–32.7)		31.0 (28.8–33.2)		30.6 (27.7–33.6)	

Results adjusted for age at diagnosis, sex, educational level, province of recruitment, stage at diagnosis and histological grade at diagnosis analysis. * Categories used as reference in the analysis. (1) PCS-12: Physical Component Summary of SF-12, (2) MCS-12: Mental Component Summary of SF-12, (3) FCSI: a (Functional Assessment of Cancer Therapy) Colorectal Symptom Index.

## Data Availability

The original contributions presented in the study are included in the article/Appendix A; further inquiries can be directed to the corresponding author.

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
