# Peer review of "Health-Related Quality of Life in Long-Term Colorectal Cancer Survivors"

_healthcare, 2024, doi:10.3390/healthcare12191917_

Round 1
Reviewer 1 Report
Comments and Suggestions for Authors
In the introduction, it should be mentioned how health-related quality of life can affect both long-term morbidity and mortality.
Justify why you have used the SF-12 (12-Item Short-Form Health Survey) and FCSI (Colorectal Symptom Index) and the novelty that this study brings compared to other studies with similar characteristics.
In a subsample of cancer patients, a prior validation should have been conducted in this population.
In the Study Design and Participants section, a brief description of the patient characteristics should be provided, including the percentage of men and women, the general distribution of the tumour (distal or proximal), and the age.
This sentence would be more appropriate for the statistical analysis section: Cronbach’s α was used to measure the internal consistency of the SF-12 and FCSI 134 found that the values was higher than the proposed standard of 0.7 (SF-12: PCS 0.85 and 135 MCS 0.78; FCSI: 0.81) [18,22].
In table 1 What does the asterisk next to the age refer to? It is not located in the table's footnotes.
Age categorisation would have been improved by considering the median age, as this would have resulted in more participants per group and greater statistical power to detect differences.
It would be interesting to define what is considered for each category of education, and if there were individuals with no education, they should also be included.
Were there any cases where a participant received two different treatments (for example, radiotherapy and chemotherapy)?
In the statistical analysis, justify with bibliographic references the reason for using the t-test to compare means without checking for normality.
Explain how you addressed the issue if any variable did not meet the requirements for linear regression.
The methodology should include details on the sample size estimation for this study.
Include in the text how the BMI was calculated and justify the ranges applied in this study. It may be a typographical error in Table 1 with the value of 17.5.
It would be interesting to apply Lasso regression models in this study to improve the precision and interpretation of the regression model. Applied Bonferroni correction .
Justify the reasons for using the different categories of the various variables in the regression models. It would be of interest to include references to support this selection.
Reviewer 2 Report
Comments and Suggestions for Authors
This is a study about the quality of life (QoL) of patients who suffered from colorectal cancer in Spain. It is found that age and gender most determine QoL, while other demographic factors, health behaviour, tumor characteristics and variables related to treatment were found to be largely unrelated to measured QoL.
This is a valuable study, even though the results are largely negative. Research question, sampling, methods and results are (mostly) clearly described. There are, however, some questions and issues which need to be resolved.
- This is a case-only study (line 164). Yet, controls are mentioned in l. 86, and MCC-Spain includes population controls. Why were these not used in the analysis? This might have resolved the ambiguity about the effect of age. It is found that QoL declines with age among the patients, but this may also well be the case within the population in general. The Discussion mentions this questoin, but using the controls might lead to a much clearer conclusion.
- The title says: "7 years after diagnosis". But section 2.1 says that patients were recruited between 2008 and 2013, while the questionnaires were administered between 2017 and 2018. This set-up seems to imply a variable time after diagnosis (which might have been included among the independent variables.
- The study does indeed cover many areas of Spain, but some regions are not represented at all (e.g. Andalucia, Extremadura), while Barcelona and also León seem overrepresented. How did this regional distribution come about?
- Tables 2 and 3 are very large, and given that most of the differences between categories are not significant, it is not clear to me that including them in the article is useful. They might be put in Supplementary material. On the other hand, Figure S1 is a clear summary of the selection process, and might be included.
- Surely, the (unavoidable) fact that only survivors could be surveyed is an important limitation of the study.
Some minor remarks:
- l. 191: "punctuation" what is this?
- l. 243: "benefit-finding". Please explain this a bit more.
- l. 267: "systematic review". No reference for this is provided, it seems.
- ll. 290-291 "turning this limitation into a strength". A 65.4% response rate is indeed not bad at ll, but to write that this is a strength is too strong.
Finally, I would like to mention an issue that is outside this review. As a social scientist, I was expecting something about the possible selection bias due to the death of many patients. This is not mentioned in the manuscript, but it seems that this is also true for most if not all articles on QoL of cancer (and other) patients. It seems unreasonable to make a recommendation that goes rather beyond what is usual in the field, and therefore, I do not ask you to take selection bias into account.
In the social sciences where I work, and in particular in labour economics, selection bias is a familiar phenomenon, and methods have been developed to deal with this. See e.g. the references below. While, as said, I do not ask you to take this up, you might consider writing another article that introduces the issue of survivor selection bias in your field of research. As this would be a significant methodological innovation, it might generate many citations.
West BT, Little RJ, Andridge RR, Boonstra PS, Ware EB, Pandit A, Alvarado-Leiton F. ASSESSING SELECTION BIAS IN REGRESSION COEFFICIENTS ESTIMATED FROM NONPROBABILITY SAMPLES WITH APPLICATIONS TO GENETICS AND DEMOGRAPHIC SURVEYS. Ann Appl Stat. 2021 Sep;15(3):1556-1581. doi: 10.1214/21-aoas1453. Epub 2021 Sep 23. PMID: 35237377; PMCID: PMC8887878. and also Selection bias - Wikipedia .
Round 2
Reviewer 1 Report
Comments and Suggestions for Authors
With the changes made and the improvements and justifications included, the article has increased in quality, and facilitates reproducibility.
Reviewer 2 Report
Comments and Suggestions for Authors
Thank you for addressing my concerns. I have no further comments.